# Presence of B19V in Patients with Thyroid Gland Disorders

**DOI:** 10.3390/medicina55120774

**Published:** 2019-12-04

**Authors:** Sabine Gravelsina, Zaiga Nora-Krukle, Simons Svirskis, Egils Cunskis, Modra Murovska

**Affiliations:** 1Institute of Microbiology and Virology, Rīga Stradiņš University, 5 Ratsupites St., LV-1067 Riga, Latvia; zaiga.nora@rsu.lv (Z.N.-K.); ssvirskis@latnet.lv (S.S.); modra.murovska@rsu.lv (M.M.); 2Department of Surgery, Riga East Clinical University Hospital “Gailezers”, 2 Hipokrata St., LV-1039 Riga, Latvia; cunskis@inbox.lv

**Keywords:** B19V, thyroid gland diseases

## Abstract

*Background and Objectives:* Viral infections are frequently cited as a major environmental factor implicated in thyroid gland diseases. This work aimed to estimate the presence of B19V infection in patients with thyroid gland disorders. *Materials and Methods:* Thyroid gland tissue and blood samples of 50 patients with autoimmune thyroid gland diseases (AITDs), 76 patients with non-autoimmune thyroid gland diseases (non-AITDs), and 35 deceased subjects whose histories did not show any autoimmune or thyroid diseases (control group) were enrolled in the study. Virus-specific IgM and IgG were detected using ELISA, and the presence and viral load of B19V in the tissue and blood were detected using PCRs. *Results:* B19V IgG antibodies were detected in 35/50 AITDs patients and in 51/76 non-AITDs patients, and B19V IgM antibodies were detected in 1/50 patients with AITDs and in none of the 76 patients with non-AITDs. The B19V NS sequence was found in the tissue DNA of 10/50 patients with AITDs, in 30/76 with non-AITDs, and in 1/35 control group individuals. The median B19V load in the tissue of patients with AITDs and non-AITDs was 423.00 copies/µg DNA (IQR: 22.50–756.8) and 43.00 copies/µg DNA (IQR: 11.50–826.5), respectively. The viral load in one of the 35 nPCR B19V-positive thyroid tissue samples from the deceased subjects was 13.82 copies/µg DNA. The viral load in the tissue of patients with AITDs was higher than in whole blood, which possibly indicates B19V persistency in thyrocytes (*p* = 0.0076). *Conclusion:* The fact that the genoprevalence of B19V NS was significantly higher in patients with non-AITDs compared to the control group and in the thyroid gland tissue of patients with AITDs, and that the non-AITDs viral load was higher than in tissue derived from the control group individuals, suggest the possibility that B19V infection could be involved in the development of thyroid gland diseases.

## 1. Introduction

The human parvovirus B19 (B19V) is a small, non-enveloped virus that belongs to the Parvoviridae family. The B19V genome is a linear single-stranded DNA and encodes one non-structural (NS) protein and two viral capsid (VP) proteins [1]. The transmission of B19V infection is through respiratory droplets. Another well-documented route of spread for B19V is vertically—from mother to fetus and from parenteral transfusion with contaminated blood products or needles [2]. IgM antibodies appear 10 days after infection and last for about three months [3], but IgG antibodies appear two weeks after infection and may persist indefinitely [4]. Erythroid progenitor cells are the main sites of B19V replication and globoside is considered to be the primary receptor of B19V [5]. B19V DNA has been reported in association with persistent infection in multiple tissues, including bone marrow, brain, colon, heart, kidneys, liver, lungs, lymphoid, skin, synovium, testis, thyroid, and tonsils [6]. Most often, B19V infection is asymptomatic or has mild, self-limiting, nonspecific, cold-like symptoms and is controlled by the development of a specific immune response, but in many cases the clinical situations can be more complex. It has been well documented that the most common manifestation of B19V infection is erythema infectiosum, or fifth disease, a rash illness of childhood characterized by a “slapped cheek” [7,8]. Besides erythema infectiosum, several more clinical conditions have been linked to the virus, such as arthropathy; transient aplastic crisis; chronic red cell aplasia; papular, purpuric eruptions on the hands and feet (“gloves and socks” syndrome); and hydrops fetalis. Conditions (diseases) postulated to have a link to parvovirus B19V infections include encephalopathy, epilepsy, meningitis, myocarditis, and dilated cardiomyopathy [9]. B19V has also been linked to autoimmune diseases such as autoimmune neutropenia, thrombocytopenia, haemolytic anaemia, and rheumatoid arthritis [10,11]. Viral infections are frequently cited as a major environmental factor involved in thyroid gland diseases [10,12]. The presence of viruses in the thyroid has been shown, but whether they are implicated in thyroid diseases or are only spectators is under investigation. During the last few years, a limited number of studies have suggested that B19V infection may be associated with autoimmune thyroiditis [13,14]. According to Mori et al., the intra-thyroidal persistence of B19V DNA in a patient with Hashimoto’s thyroiditis has been detected. The cell types responsible for the B19V DNA persistence are not determined and the immune cells infiltrating the thyroid gland may be the source of the B19V DNA. However, the possibility that thyroid epithelial cells harbor B19V DNA cannot be excluded [15].

## 2. Materials and Methods

### 2.1. Samples

Fresh frozen peripheral blood and thyroid tissue samples from 126 patients, including 50 with different types of autoimmune thyroid diseases (Hashimoto thyroiditis and Grave’s disease) and 76 patients with non-autoimmune diseases (papillary thyroid carcinoma, adenoma, and goiter), who underwent thyroidectomy between December 2010 and June 2015 at the Department of Riga East Clinical University Hospital, Clinic “Gaiļezers” were enrolled in the study. The age of the patients ranged from 25 to 78 years (median age 52 years (IQR: 41–61) and only 12 of them were men. The exact diagnosis of the thyroid gland pathology was established by histological analysis of the resected tissue and analysis of the specific autoantibodies (auto-antibodies against thyroid peroxidase (TPO), thyroglobulin (TG), and thyroid stimulating hormone receptor (TSHr)).

Whole blood and thyroid tissue samples from 35 randomly selected age and gender matched deceased subjects without thyroid pathologies (the deceased subject’s history did not show any autoimmune or thyroid diseases) who had been autopsied were identified as being eligible for the control group. The determined post-mortem interval was between 7 and 30 h and the material was stored at −80 °C. The study design was approved by the Ethical Committee of Rīga Stradiņš University (ethical code Nr 67 and date of approval 25 October 2012), and written consent was obtained from all the patients and relatives, respectively.

### 2.2. B19V Serology by ELISA

To evaluate the seroprevalence of B19V infection among patients with thyroid gland diseases, all the plasma samples were tested for the presence of B19V-specific IgG and IgM antibodies using the Recomwell Parvovirus B19V enzyme immunoassay kit from Mikrogen Diagnostik (Neuried, Germany).

### 2.3. Nested Polymerase Chain Reaction

DNA from the thyroid gland tissue and whole blood samples was isolated by proteinase K digestion followed by phenol-chloroform extraction and ethanol precipitation, and finally, it was re-suspended into molecular-grade water. The quantity of the DNA was measured spectrophotometrically. To ensure the quality of the DNA, a β-globin PCR was performed [16]. The extracted DNA (~200 ng/µL) was used as a template in nPCR, using primers amplifying the NS gene sequence as described previously [17]. The amplified DNA with the expected size (103 bp) was analyzed in a 1.7% agarose gel. The sensitivity of the B19-specific primers was 1–10 copies per reaction [17]. As a positive control, the purified plasmid vector encoding a specific cloned fragment from Parvovirus B19V genome (AMPLIRUN^®^ Parvovirus B19V (plasmid) DNA Control; Granada, Spain) was used. DNA samples that were negative for B19V-specific sequences (DNA obtained from practically healthy B19V-negative blood donors) and water controls were included in each experiment to exclude the possibility of contamination during PCR. 

### 2.4. Quantitative Real-Time PCR

nPCR-positive B19V DNA samples were also analyzed with quantitative real-time PCR using a ViroReal Parvovirus B19V kit from Ingenetix (Vienna, Austria) and was performed on a CFX 96 touch real-time PCR detection system from Bio-Rad. The target area of qPCR was the VP1 gene of the Parvovirus B19 genotypes 1, 2, and 3. The smallest amount of template that could be detected with this kit was five template copies per PCR and this was used as a threshold in future analysis (>5 copies/µg DNA and <5 copies/µg DNA per sample).

### 2.5. Stastitical Analysis

All the graphs, calculations, and statistical analyses were performed using the GraphPad Prism software version 8.2.1 for Mac (GraphPad Software, San Diego, CA, USA). The normality of the numerical data was determined by the D’Agostino and Pearson, and Shapiro–Wilk normality tests, and if the data was not normally distributed, the non-parametric Kruskal–Wallis (KW) test followed by the two-stage step-up method of Benjamini, Krieger, and Yekutieli as the post-hoc procedure was applied. By qualitative analysis of the B19V infection rates in the subjects studied, the difference between the groups was assessed using the Chi-square (Chi2) test, to compare the proportions of respective cases. The mean levels of the parameters were expressed as medians with dispersion, characterized by the interquartile region (IQR), and a *p*-value less than 0.05 (*p* < 0.05) was considered as a statistically significant difference.

## 3. Results

### 3.1. B19V Serology by ELISA

Specific anti-B19 IgG antibodies were detected in 35 (70%) out of 50 patients with autoimmune thyroid gland diseases (AITDs) and very similar rates were detected in the group of patients with non-autoimmune thyroid gland diseases (non-AITDs)—51 (67.1%) out of 76 patients, without a statistically significant difference between the groups (*p* = 0.8454). None of the 76 patients with non-AITDs was positive for B19V IgM, while among patients with AITDs, one had virus-specific IgM and IgG simultaneously.

### 3.2. B19V NS Detection by Nested Polymerase Chain Reaction

All the DNA samples were positive for β-globin PCR and were therefore eligible for further study. The B19V genomic sequence was found in blood and/or thyroid tissue DNA samples in 14 out of 50 patients with AITDs (Figure 1)—in 9 (64.3%) patients in thyroid gland tissue DNA samples only, in 4 (28.6%) patients in blood DNA samples only, in 1 (7.1%) patient in both the blood and tissue DNA samples. The B19V genomic sequence was detected in 35 out of 76 blood and/or thyroid tissue DNA samples from patients with non-AITDs (Figure 1)—in 25 (71.4%) patients in the thyroid gland tissue DNA samples only, in 5 (14.3%) patients in the blood DNA samples only, and in 5 (14.3%) patients in both the blood and tissue DNA samples. In turn, the B19V genomic sequence was found in 5 out of 35 DNA samples derived from deceased subjects (Figure 1)—in 1 case (2.9%) only in the thyroid tissue DNA sample and in 4 cases (11.4%) in the blood DNA samples.

### 3.3. B19V Load by Quantitative Real-Time PCR

Tissue samples of all the patients and control group individuals who were B19V-positive by nPCR were also analyzed for viral load. The results showed that the median B19V load in tissue from patients with AITDs and non-AITDs was 423.00 copies/µg DNA (IQR: 22.50–756.8) and 43.00 copies/µg DNA (IQR: 11.50–826.5), respectively. The median viral load in blood from patients with AITDs and non-AITDs was 3.00 copies/µg DNA (IQR: 2.50–4.00) and 22.57 copies/µg DNA (IQR: 3.00–805.3), respectively. The data of the B19V load comparative analysis between patients with AITDs and non-AITDs showed that there was a significantly higher viral load in tissue compared to in blood of patients with autoimmune thyroid diseases (*p* = 0.0076; KW) (Figure 2). The viral load in the one of the 35 nPCR B19V-positive thyroid tissue samples from the deceased subjects was 13.82 copies/µg DNA. In the whole blood of two individuals, it was less than <5 copies/µg DNA (samples of additional two individuals were not tested due to the lack of material).

## 4. Discussion

Despite the fact that B19V was discovered in 1974, it still offers a continuous challenge to virologists. Despite a great deal of effort to understand the nature of virus-associated thyroid gland diseases, the processes that underlie the progression from viral infection to an autoimmune disease and, finally, to thyroid failure, remain poorly understood. Specific anti-B19V IgG class antibodies were found in 70% (35/50) of the plasma samples from patients with AITDs and almost with the same frequency—67.1% (51/76) in plasma samples derived from the patients with non-AITDs. These results are in the line with the results of a previous study, which also demonstrated high prevalence (75%; 48 out of 64) of specific anti-B19V IgG class antibodies in patients with AITDs [18]. In a previous paper of ours, we also analyzed the B19V seroprevalence in practically healthy blood donors where anti-B19V IgG class antibodies were found in 49%, IgM in 2%, and both IgM and IgG simultaneously in 10% of all 90 analyzed cases [19]. There was only one IgM- and IgG-positive patient (40 years old women) out of all the 126 patients with different types of thyroid disorders in this study. That patient’s blood sample was also DNA-positive. Surprisingly, we did not find the B19V genomic sequence in the thyroid tissue DNA sample in that B19 IgM- and IgG-positive patient. In some cases, collection of the blood sample might have been too early to detect a positive B19V IgM response, for example in cases of exanthema [20]. However, it is reasonable to think that in the case of patients with thyroid diseases, blood samples may have also been collected too late to detect IgM, because patients do not undergo thyroidectomy on the day of which the thyroid gland disorder was diagnosed. nPCR is an appropriate and sensitive method for the detection of B19V genomic sequences in DNA samples, and it is the beginning of understanding the relationship between B19V infection and thyroid gland diseases. B19V genomic sequence detection in tissue DNA samples matched to IgG in plasma samples can confirm that the detected viral genomic sequences are from persistent infection, and not acute. This is the first time in which the control group’s tissue was taken from deceased individuals whose histories did not show of any autoimmune or thyroid diseases, and their tissue material had been analyzed for B19V presence. Our results clearly show that the presence of B19V DNA can be found in the thyroid gland independent of whether there was or wasn’t an underlying autoimmune disease and was observed to be more frequent in not-AITDs patients—20% and 40%, respectively; *p* = 0.0613. This is in line with the published article by Adamson et al., where they strongly supported that B19V can infect normal and also cancerous thyroid tissues [21]. On the other hand, the fact that the B19V genomic sequence was found to be statistically more frequent in patients with AITDs and non-AITDs tissue DNA samples (*p* = 0.0231 and 0.0001, respectively) than in the control group individuals whose histories did not show of any autoimmune or thyroid diseases, allows us to think that B19V is not only just a bystander, but could be implicated in thyroid diseases. Wang et al. detected a higher prevalence (89.6%) of the B19V DNA in thyroid carcinomas and a lower prevalence (43.8%) in the controls [22]. Later Wang and colleagues reported that the B19V DNA is also found in the thyroid tissue of 96.7% (29/32) of adults with Hashimotos’s thyroiditis and in 44% (7/16) of normal thyroids [14]. In this study, the viral load was higher in the tissue than in whole blood, which possibly indicates the persistency of B19V in thyrocytes (*p* = 0.0076). The cell types responsible for the B19V persistence are not analyzed in our study, though immune cells infiltrating the thyroid gland might be the source of the B19V DNA. However based on a few previous studies the possibility that thyroid epithelial cells harbor the B19V DNA cannot be excluded [6] 

## 5. Conclusions

The fact that the genoprevalence of B19V NS was significantly higher in patients with non-AITDs comparing to the control group, and that the viral load in the thyroid gland tissue of patients with AITDs and non-AITDs was higher than in the tissue derived from the control group individuals, suggest that B19V infection could be involved in the development of thyroid gland diseases.

## Figures and Tables

**Figure 1 medicina-55-00774-f001:**
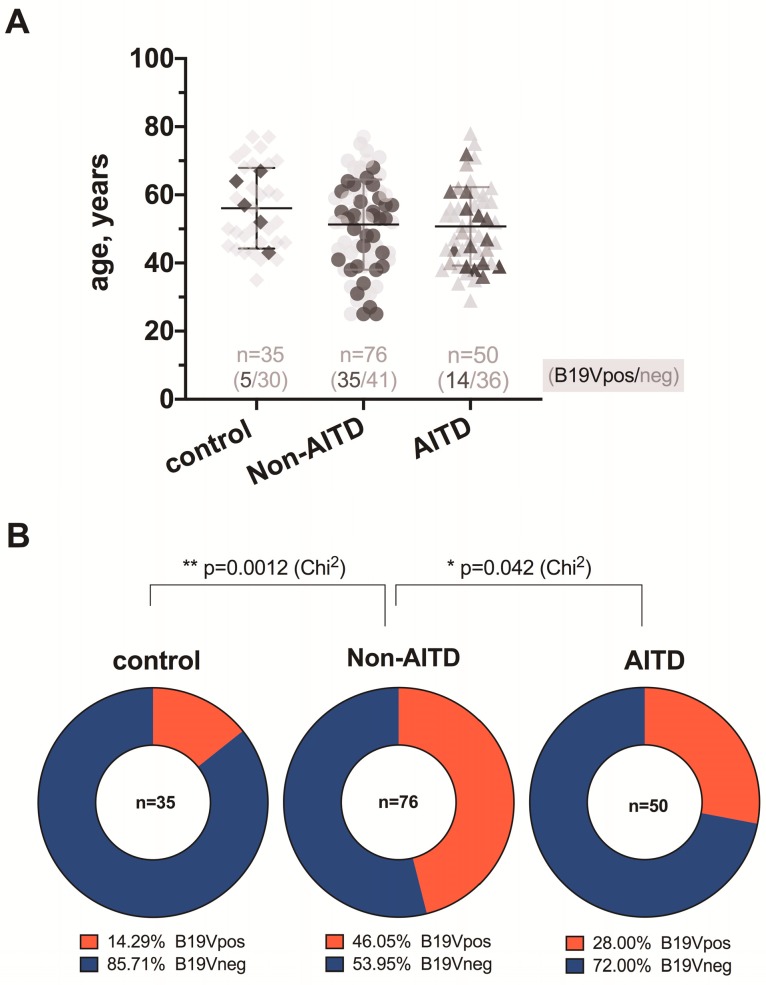
Age and B19V infection rates of patients with non-autoimmune thyroid gland diseases (non-AITDs) and autoimmune thyroid gland diseases (AITDs), and deceased subjects as control; (**A**) dark symbols represent individuals with positive B19V infection (B19Vpos), and light grey symbols represent individuals without B19V infection (B19Vneg); the corresponding B19Vpos/neg ratio of each group is represented above the *x*-axis; (**B**) between-group comparisons of the ratios and significance of differences evaluated by Chi^2^.

**Figure 2 medicina-55-00774-f002:**
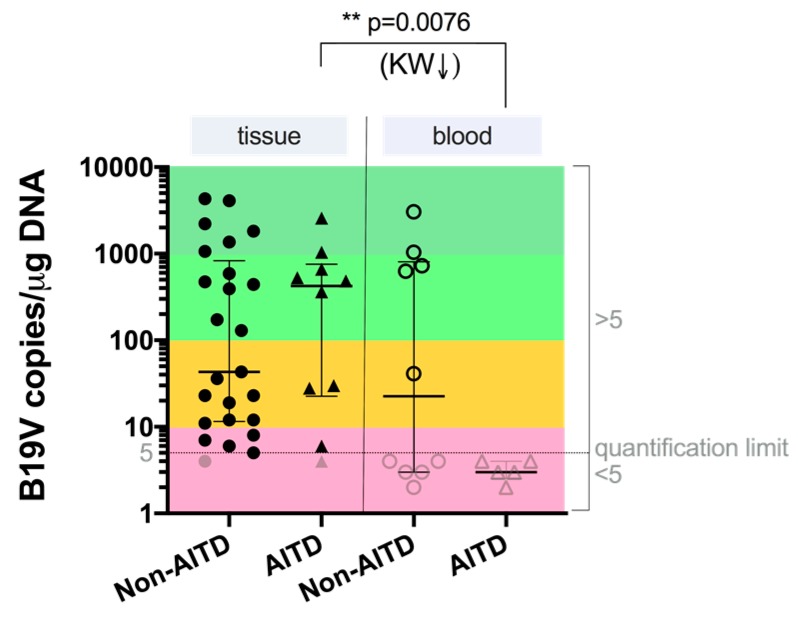
Comparison of numbers of B19V copies in the tissue and blood of patients with AITDs and non-AITDs. Light gray symbols show values under the quantification limit. Significance of differences was established using the Kruskal-Wallis (KW) test.

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
