# Peer review of "Presence of B19V in Patients with Thyroid Gland Disorders"

_medicina, 2019, doi:10.3390/medicina55120774_

Round 1

Reviewer 1 Report

Nice study on B19V in (autoimmune) thyroid diesease.

Unfortunately, this has been described before, and I cannot see any new aspects.

Author Response

Dear Reviewer,

Thank you very much for careful reviewing of our manuscript.

Most studies focus on autoimmune thyroiditis, but this study investigates the role of B19V in both autoimmune and non-autoimmune thyroiditis, and also examines tissue samples (after thyroidectomy), not just blood or plasma. The results of this study show not only the presence of the virus in the particular samples but also the viral load in the tissues, both in autoimmune and non-autoimmune thyroiditis, which is significantly different from the control group included in the study where deceased subject’s histories did not show any of autoimmune or thyroid diseases.

Reviewer 2 Report

virus implication is a subject wich needs investigation 

this publication is interesting in terms of virus load as data about BV 19 seropositivity and ADN presence is already known

Fig 1 comments should be clarified about the age : it's important to have the age of patients as seropositivity increases with age but it's not reallu explained in the text

the text is clear and concise

Author Response

Dear Reviewer,

Thank you very much for careful reviewing of our manuscript. 

The figure as well as the explanatory comments bellow the figure have been changed. 

Reviewer 3 Report

The authors report on the prevalence of parvovirus B19 in thyroid tissue in individuals with autoimmune and non-autoimmune thyroid gland diseases. The higher B19V genoprevalence in individuals with thyroid conditions in comparison to the control group leads them to conclude for a possible involvement of B19V in the etiology of thyroid pathology.

Questions and considerations:

Please specify whether the samples are fresh frozen or FFPE? This is important as crosslinking and DNA fragmentation can certainly affect the PCR results and conclusions. If FFPE samples were used this should be discussed. Also, in relation to the DNA fragmentation, what is the post-mortem time of the control samples? Were they also FFPE? The author´s report on genoprevalence in blood of recently deceased individuals, did they do serology as well? In line with the fragmentation level of the samples, many aspects in the Methods section need to be described more in detail as well as referenced properly. Refs #16 and #17 do not contain any appropriate/relevant method description. What is the length of the amplicon of the nPCR and qPCRs? What is the target area of qPCR? The authors report on performing a B globin assay, yet there is no proper reference to the method and no mention of it in the results nor in the discussion. Given that there is no report on the sensitivity of the nPCR and given the analytical resolution of a gel, why did the authors choose to do qPCR only from those samples that were positive by nPCR? When reporting viral load, in several places are reported copies/ug DNS (also in fig2). What does DNS stand for? Is it DNA? Please correct or spell out as appropriate. Could the authors perform an analysis of the fragment length distribution of the sample´s genomic DNA to strengthen their results and conclusions? The seroprevalence in the IATD group is significantly higher than the genoprevalence, calling again for a need to support the soundness of the results by PCR. Again, if the b globin was performed with this purpose, the method should be properly described or at least correctly referenced, and the results should be reported along with a discussion on the measures that the authors took to assess the quality of the DNA. The follicular composition of the thyroid tissue may benefit from collagenase as described in Pyöriä et al Nat comms 2017. Perhaps the authors could discuss this? The authors report on viral DNA in the blood of several individuals. Was there any follow up to the viremia? Any informantion on the clinical background of this patients other than the thyroid pathology? At last, given the higher genoprevalence in the cases group, the authors suggest a possible involvement of B19V in the etiology of thyroid pathology. However, the control group is too small to make any significant conclusions. In ideal conditions the ratio between cases: controls should be of 1:4, or at least of 1:1 which is clearly not the case in the present study.

Author Response

Dear Reviewer,

Thank you very much for the careful reviewing of our manuscript.  Please find the answers to your comments below:

None of the samples were FFPE, all peripheral blood and thyroid tissue samples of the patients were fresh frozen. We added this information in the manuscript. Also post-mortem blood and tissue control samples were not FFPE, they were received within 7 - 30 h from autopsy and were stored at -80oC-. We did not do serology on deceased individuals due to impossibility to separate plasma or serum from blood samples of deceased subjects Mentioned aspects in the Methods section are described more in details as well as proper references added. a) The length of nPCR amplicon is 103bp. For qPCR we used commercial kit (ViroReal Parvovirus B19V kit from Ingenetix ) and the amplicon size is not mentioned/showed by the manufactures. b) The target area of qPCR is VP1 gene of the Parvovirus B19 genotypes 1,2,3 and information is added it the text. c) My apologies for given not proper reference, it is corrected in the results section of the manuscript. d) Taking into account that the sensitivity of the nPCR used in our experiments was 1 – 10 copies per reaction (Barah et al. 2001) and sensitivity for qPCR was 5 template copies per reaction (within the sensitivity range of the nPCR) we have chosen to do qPCR only for the samples positive in nPCR. e) Sorry for spelling mistake – should be “DNA” (“DNS” in Latvian). f) Fragment length distribution of the samples genomic DNA is not done in this study. We thank you for the suggestion and will take it into account in the continuation of the project. We agree with your suggestion – for nPCR results as well as β-globin PCR are now corrected with proper reference.   We agree that composition of thyroid tissue may benefit from collagenase as described in Pyöriä and colleagues where they found that the B19V-DNA loads were significantly higher in the CD19 + cells released by collagenase. It is a good suggestion for our future experiments in this project. There was no follow up to the viremia. Peripheral blood and tissue samples were taken only once – on the day patients underwent thyroidectomy.The disease was confirmed by the clinical manifestations, complemented by thyroid biochemical tests, ultrasound imaging, and histopathological examination. 76 patients with AITD, 50 patients with non-AITD and 35 controls were enrolled in this study. This ratio between the study groups is not perfect but the differences in genoprevalence of B19V and viral loads between the groups allows to conclude that B19V infection could be involved in the development of thyroid gland diseases. We have modified the conclusion as follows :

The fact that genoprevalence of B19V NS was significantly higher in patients with non-AITD comparing to the control group, and in thyroid gland tissue of patients with AITD and non-AITD viral load was higher than in tissue derived from control group individuals allow suggest that possibly B19V infection could be involved  in the development of thyroid gland diseases.

Round 2

Reviewer 3 Report

The authors have answered all my questions.